# Decoding the 5′ nucleotide bias of PIWI-interacting RNAs

Chad B. Stein [1,6], Pavol Genzor [1], Sanga Mitra[1], Alexandra R. Elchert[1], Jonathan J. Ipsaro [2,3], Leif Benner[1,4], Sushil Sobti[1], Yijun Su[5], Molly Hammell[3], Leemor Joshua-Tor [2,3] & Astrid D. Haase [1]

PIWI-interacting RNAs (piRNAs) are at the center of a small RNA-based immune system that defends genomes against the deleterious action of mobile genetic elements (transposons). PiRNAs are highly variable in sequence with extensive targeting potential. Their diversity is restricted by their preference to start with a Uridine (U) at the 5′ most position (1U-bias), a bias that remains poorly understood. Here we uncover that the 1U-bias of Piwi-piRNAs is established by consecutive discrimination against all nucleotides but U, first during piRNA biogenesis and then upon interaction with Piwi's specificity loop. Sequence preferences during piRNA processing also restrict U across the piRNA body with the potential to directly impact target recognition. Overall, the uncovered signatures could modulate specificity and efficacy of piRNA-mediated transposon restriction, and provide a substrate for purifying selection in the ongoing arms race between genomes and their mobile parasites.

[1] National Institute of Diabetes and Digestive and Kidney Diseases, National Institutes of Health, Bethesda, MD 20892, USA. [2] W.M. Keck Structural Biology Laboratory, Howard Hughes Medical Institute, Cold Spring Harbor 11724, USA. [3] Cold Spring Harbor Laboratory, Cold Spring Harbor, NY 11724, USA. [4] Department of Biology, Johns Hopkins University, Baltimore, MD 21218, USA. [5] National Institute of Biomedical Imaging and Bioengineering, National Institutes of Health, Bethesda, MD 20892, USA. [6] Present address: PhD Program in Biological and Biomedical Sciences, Harvard Medical School, Boston MA 02115, USA. These authors contributed equally: Chad B. Stein, Pavol Genzor, Sanga Mitra, Alexandra R. Elchert. Correspondence and requests for materials should be addressed to A.D.H. (email: astrid.haase@nih.gov)

PIWI-interacting RNAs (piRNAs) and their PIWI protein partners establish restriction of transposons in germ cells, and thus guard genomic identity[1–4]. Mutations in core piRNA pathway genes result in sterility and threaten the survival of a species. While mechanisms of piRNA biogenesis differ greatly from those of other small silencing RNAs, core concepts of small RNA-guided regulation of gene expression are conserved: At the heart of all RNA silencing mechanisms resides the RNA-induced silencing complex (RISC), which consists of a small non-coding RNA and its Argonaute protein partner (AGOs and PIWIs)[5]. Within RISC, the small RNA determines target specificity by complementary base-pairing, and the Argoanute protein initiates transcriptional or post-transcriptional silencing mechanisms[6,7].

It has long been observed that several classes of small RNAs preferentially harbor a Uridine (U) at their 5′-most position, particularly piRNAs[8]. In contrast to the well-defined mechanism of microRNA (miRNA) biogenesis, processing of piRNAs from long single-stranded transcripts is poorly understood[9]. The 5′ ends of mature piRNAs are generated first, either by the action of the conserved endonuclease Zucchini(Zuc)/PLD6[10,11] or by the slicer activity of a PIWI protein itself[8,12,13]. After association with PIWI proteins, the 3′ end of mature piRNAs is generated either by a second Zuc/PLD6-cut[14], or by exonucleolytic trimming to resemble what is believed to be the footprint of the associated PIWI protein[14–16]. The resulting piRNAs are highly diverse in sequence and variable in length, and are best defined by the association with their PIWI partners.

While the first base of piRNAs is hidden within PIWI proteins, all other nucleotides could contribute to target recognition. The rules of their target engagement, however, remain largely elusive, and potentially range from fully complementary base-pairing to a seed-based recognition mechanism[17,18]. Together with the high diversity of piRNAs, with hundreds of thousands of unique sequences, and lack of sequence conservation, it remains unclear how target-specificity of these potent silencing pathways is regulated[4,8,12,19,20].

One prominent feature restricts the enormous sequence space of mature piRNAs: they preferentially harbor a Uridine (U) at their 5′ most position. This 1U-bias is conserved across species and has also been observed for other classes of small RNAs[9]. While the molecular source of the 1U-bias and its function are poorly understood, the physical position of the small RNA's first base within RISC is well-defined as anchored to a specialized pocket in the middle (MID) domain of Argonaute proteins, termed the specificity loop (SL)[21]. Based on structural data and in vitro binding studies, the SL has been proposed to establish the 1U bias by selecting for 1U small RNAs[16,22–25]. However, the recent observation of phased piRNA production by the endonuclease Zucchini (Zuc) implies 1U-specific processing of piRNA precursors instead[26,27]. Here, we investigate both hypotheses in vivo, using Drosophila as a model system. Our data support that the 1U-bias of Piwi-piRNAs is established by the consecutive and differential selection against all nucleotides but U, first during piRNA biogenesis and then by Piwi's specificity loop. Furthermore, we uncover processing-dependent selection against U within the piRNA body with the potential to directly shape the piRNA-target repertoire. Overall, we propose that the complex establishment of an ultimate 1U-bias could provide a substrate for purifying selection to improve the specificity and efficiency of transposon silencing.

## Results

### The specificity loop (SL) contributes to but does not solely determine the 1U-bias. To directly probe the impact of the SL on

the 1U-bias of piRNAs, we generated mutants for Piwi's SL, characterized their associated piRNAs, and investigated their function in Drosophila ovaries and in ovarian somatic sheath cells (OSC)[28,29]. Our Piwi-SL mutants comprise three categories by design (Fig. 1a, b): (I) Substitute mutants exchange Piwi's SL with the SL of other Argonaute proteins, whose physiological first nucleotide bias is well defined, including a 1C-bias (At AGO5)[22,24] and equal first nucleotide distribution (Dm Ago3)[8,13]. (II) Synthetic loops were designed to weaken the 1U-fit based on available protein structures[21,30–34], including a complete replacement of Piwi's SL with a flexible stretch of three Glycine-Serine-Serine repeats (GSS). Finally, we generated a loop deficient (LD) mutation (III) that removes the SL entirely. We purified FLAG-HA-tagged Piwi mutants (FH-Piwi-SL) from OSC and analyzed their associated small RNAs by Illumina sequencing (Fig. 1c). PiRNAs associated with Piwi-SL mutants were similar to wild type in length, targeting potential, and genomic annotation (Fig. 1d–f, Supplementary Fig. 1a, b). Surprisingly, while most alterations to Piwi's SL slightly reduced the 1U frequency of Piwi-piRNAs, none of the mutants abrogated the observed 1U-bias, thus suggesting an SL-independent establishment (Fig. 1g, Supplementary Fig. 1c).

**The identity of Piwi's SL is not required for Piwi's function in vivo.** To re-evaluate this observation in vivo, and to investigate the biological impact of alterations to Piwi's SL, we characterized Piwi-SL mutants in Drosophila ovaries. FH-Piwi-SL(Ago3) and FH-Piwi-SL(GSS), restored fertility and ovary morphology in a piwi null background similar to a wild type rescue construct, but in contrast to the loop-deficient mutant (Piwi-LD) (Fig. 2a, and Supplementary Fig. 2a–c). The potential of Piwi-SL-mutants to restore piwi function correlated with their ability to form functional Piwi-piRISC, as indicated by the accumulation of nuclear FH-Piwi-SL protein[35,36] (Fig. 2b). Thus, a particular primary sequence of Piwi's SL is not required for Piwi function in vivo, but complete removal of the SL likely impairs protein folding and consequently piRISC formation as previously described[36,37]. PiRNAs associated with Piwi-LD (in piwi > Gal4 heterozygous flies (indicated by *)), however, closely resemble those associated with other SL mutants and Piwi wt in length (Fig. 2c), targeting potential (Fig. 2d), and genomic annotation (Fig. 2e). As in OSC, Piwi-piRNAs maintained a 1U-bias in SL mutants with 1U frequencies above 60% in all experiments (Fig. 2f). Taken together, our comprehensive characterization of Piwi-SL mutants revealed that the 1U-bias of Piwi-piRNAs is established independently of but reinforced by Piwi's SL. Our SL mutants eliminate SL-dependent nucleotide selection in the first position but retain the ability to associate with Zuc-dependent piRNAs. Thus, they likely grant an unrestricted view on piRNA 5′-termini generated by the Zuc-processor complex, which are otherwise rapidly degraded in the absence of their Piwi protein partner[38,39].

**Preferences during piRNA biogenesis establish sequence restrictions.** To characterize potential processing preferences that could extend beyond the first position of mature piRNAs in the context of the precursor transcripts, we collapsed all uniquely mapping piRNAs by their 5′ end (position 0), extended their genomic interval, and calculated nucleotide frequencies for each position within a 101 nt window (Fig. 3). This metagene analysis revealed that the U-rich mature 5′ ends of Piwi-piRNAs reside in a relatively U-poor and G-rich environment, which spans the observed piRNA body (0 to +25; indicated in gray) and one piRNA length upstream (−1 to −25) (Fig. 3a). We also observed peaks in U frequencies and reciprocal drops in purine frequencies (A and G) one piRNA-length upstream (−26) and downstream (+26) of the observed piRNA 5′ end, perhaps indicating the first

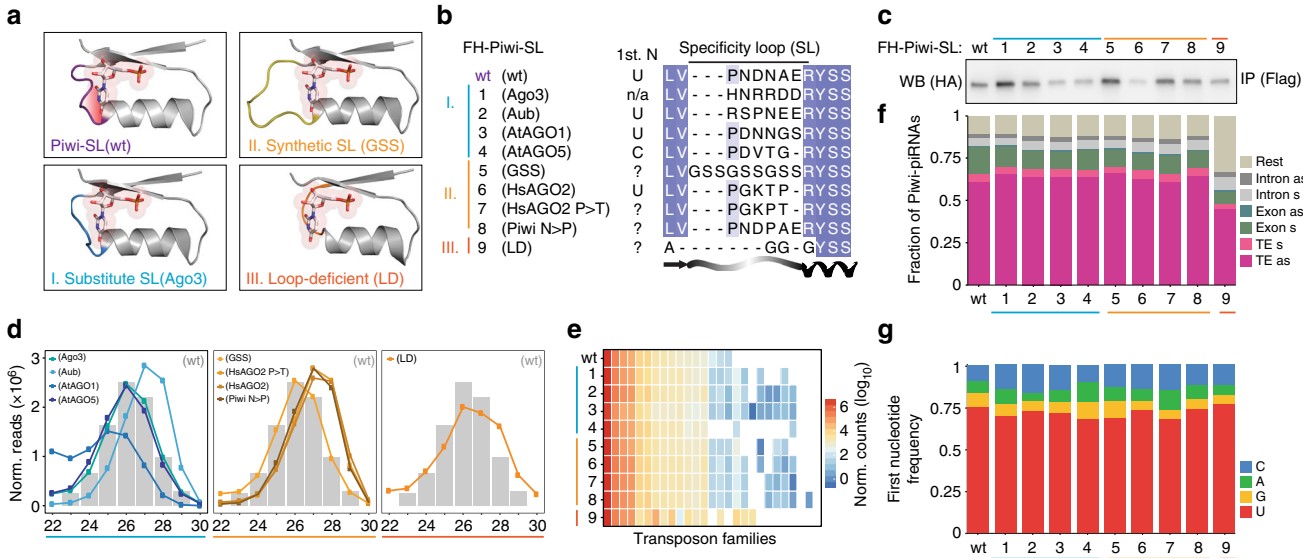

**Fig. 1** The Specificity Loop (SL) contributes to, but does not solely determine the 1U bias of Piwi-piRNAs in ovarian somatic sheath cells (OSC). **a** Structural model of Piwi's SL and of SL mutants in complex with monophosphorylated Uridine (U) based on the crystal structure of SIWI (PDB 5GUH). The SL is highlighted in purple for wild type (wt), blue for the substitute SL from Dm Ago3, yellow for the synthetic 3× (GSS) loop, and orange for the GGG linker in the loop deficient (LD) mutant. The recognition interface between the SL and the Uracil nucleobase is indicated in red for wt. **b** Flag-HA (FH) tagged Piwi constructs harboring wild type (wt) or mutant specificity loops (SL) (FH-Piwi-SL). The first nucleotide preference of small RNAs that are physiologically associated with the original Argonaute protein is indicated (1st N). **c** FH-Piwi-SL proteins were expressed in OSC, immunoprecipitated (IP) using anti-FLAG antibody, and detected by western blotting (WB) by anti-HA antibody. **d** Nucleotide (22–30nt) length distribution of FH-Piwi-SL-piRNAs that associate with wt (gray) and the three categories of SL mutants. **e** Annotation of FH-Piwi-SL-piRNAs by their antisense targeting-potential for different transposon families. **f** Genomic origin of FH-Piwi-SL-piRNAs with respect to annotation categories in sense (s) and antisense (as) orientation. **g** Nucleotide frequencies at the first position of FH-Piwi-SL-piRNAs. [$M \leq 100$: Unique mapping reads and primary alignments of multimapping reads are considered for the analyses represented in **d–g**; read length was restricted to 24–29nt for analyses presented in **e–g**]. (See also Supplementary Fig. 1 and Supplementary Data 3.)

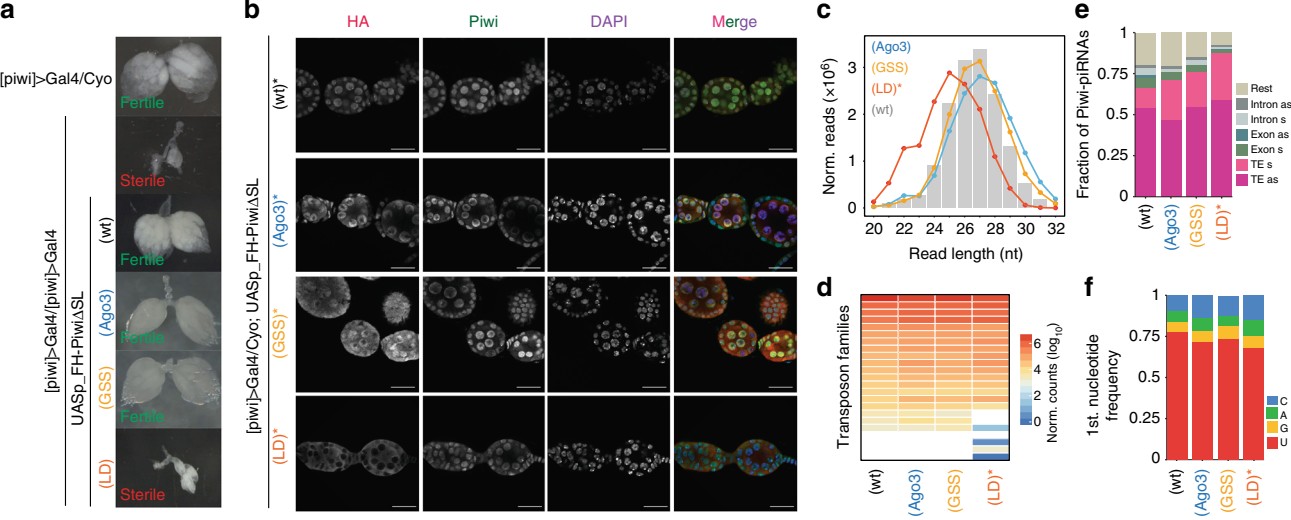

**Fig. 2** The identity of Piwi's SL is not required for Piwi's function in vivo. **a** Flies expressing Gal4 instead of Piwi from the *piwi*-promoter ([piwi]>Gal4) are heterozygous fertile, but homozygous sterile. A Gal4-inducible Piwi transgene (UASp_FH-Piwi(wt)) and select SL mutants rescue fertility and ovarian morphology. **b** Localization of FH-Piwi-SL transgenes (IF: anti-HA) and all Piwi (IF: anti-Piwi) in piwi heterozygous (*) ovaries. **c** Nucleotide (nt) size distribution of piRNAs that associate with FH-Piwi wt (gray) and SL mutants. [PiRNAs associated with FH-Piwi-SL(wt), FH-Piwi-SL(Ago3), and FH-Piwi-SL (GSS) were purified from rescued piwi null ovaries. FH-Piwi-SL(LD)-piRNAs were purified from piwi heterozygous flies (*)]. Scale bars indicate 20 μm. **d** Annotation of piRNAs by their potential to target different transposon families by antisense complementarity. **e** Genomic origin of piRNAs with respect to annotation categories in sense (s) and antisense (as) orientation. **f** Nucleotide frequencies at the first position of FH-Piwi-SL piRNAs. [$M \leq 100$: Unique mapping reads and primary alignments of multimapping reads are considered for the analyses represented in **d–f**; read length was restricted to 24–29 nt for analyses presented in **d–f**.] (See also Supplementary Fig. 2 and Supplementary Data 4.)

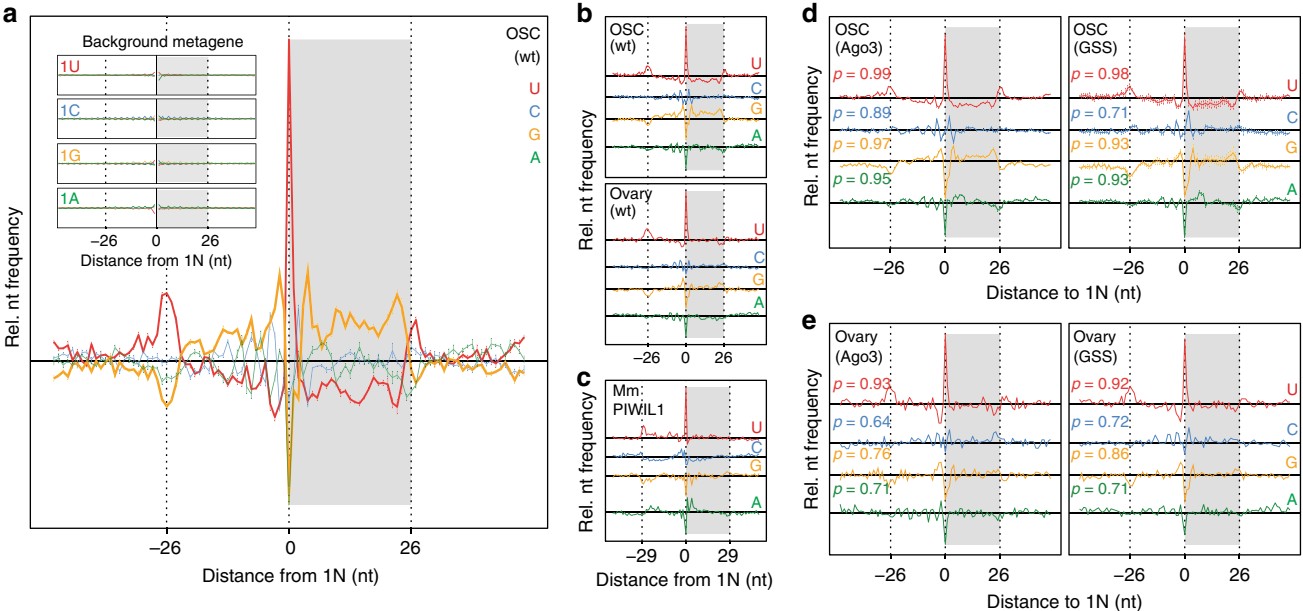

**Fig. 3** Preferences during piRNA biogenesis establish sequence restrictions. **a** Patterns of nucleotide frequencies within a 101 nt window surrounding the 5′ end of Piwi-piRNAs (position 0) in OSC. Data represent the mean of three biological replicates ($n = 3$) and standard deviations are indicated. Nucleotide 1–26 of the observed piRNAs are indicated by a gray box. The insert illustrates the background model with all possible 1U, 1C, 1G, or 1A starting piRNAs from Piwi-piRNA clusters. **b** Patterns of nucleotide frequencies for all four nucleotides of Piwi-piRNAs from OSC, fly ovaries, and **c** mouse Piwil1-associated piRNAs. Patterns of nucleotide frequencies of piRNAs associated with Piwi-SL mutants in OSC ($n = 3$) (**d**), and fly ovaries ($n = 1$) (**e**). Correlation coefficient (Pearson) compared to Piwi-SL-wt are indicated (**d**, **e**). [$M = 1$: Only unique mapping reads are considered to ensure unambiguous genomic coordinates. (See also Supplementary Fig. 3 and Supplementary Data 5.)

position of neighboring piRNAs. This pattern is reminiscent of a phased processing signature[26,27,40], and likely represents coupled processing of the 5′ and 3′ termini of neighboring piRNAs by a single Zuc-cleavage event. The observed sequence preferences are dependent on piRNA processing as they do not reflect a bias in the genomic sequence of piRNA clusters when considering all possible 5′ positions, stratified by the first nucleotide, in our background analysis (Fig. 3a insert). These signatures were consistent between Piwi-piRNAs from OSC and from fly ovaries, and to a great extent conserved in Miwi/Piwil1-piRNAs from adult mouse testis (Fig. 3c, d). Importantly, these preferences were independent of Piwi's SL with correlation coefficients (Pearson) for individual nucleotides greater than 0.6 across the 101 nt interval in OSC and *Drosophila* ovaries (Fig. 3d, e, and Supplementary Fig. 3). While the 5′-most nucleotide of piRNAs is deeply buried in PIWI's MID domain and not available for target recognition by base pair complementarity, the 1U-bias has the potential to modulate the overall target space by altering the representation of individual piRNA sequences. In contrast, the observed preferences throughout the piRNA body directly shape the piRNA-target interface.

**The 1U-bias is established by a universal mechanism across piRNA generating regions**. Most piRNAs originate from a few piRNA generating regions termed piRNA clusters[8,12]. If the 1U-bias is indeed primarily established by a general piRNA processing mechanism, we would predict that the 1U frequency of piRNA-groups, according to their generating clusters, should correlate with the T-content of the cluster itself and result in a constant 1U-enrichment factor. To test this hypothesis, we correlated the first nucleotide frequencies of piRNA-groups with the genomic nucleotide frequencies of the corresponding clusters (Fig. 4a). The observed 1U frequencies of piRNA-groups positively correlated with the genomic T-content of their generating

clusters (Correlation coefficient (Pearson) $p > 0.5$) with a universal enrichment of about 2.3-fold (observed over expected 1U frequencies) (Fig. 4a insert), further supporting 1U enrichment during piRNA biogenesis.

Our metagene analysis as well as characterization of individual piRNA cluster-groups suggest the establishment of a 1U-bias during piRNA processing. These findings offer an explanation why Ago3-piRNAs exhibit an unexpected 1U-bias when aberrantly fueled by the Zuc-processor instead of ping-pong-generated piRNAs in OSC[41]. Nevertheless, the observed processing-dependent sequence preferences are surprising, because neither fly nor mouse Zuc exhibit obvious sequence-specific processing in vitro[10,11]. However, cofactors or post-translational modifications likely create sequence preferences of the Zuc-processor complex in vivo[26,42].

**Piwi's SL selects against 1C-piRNAs to re-enforce an overall 1U-bias**. Interestingly, and reinforcing our interest on Piwi's SL, the most robust SL-dependent change affected 1C-piRNAs across different clusters in OSC (Fig. 4b) and in flies (Supplementary Fig. 4a, b), despite the overall U-richness and A-richness of these piRNA generating regions (Supplementary Fig. 4c). The observed ~40% increase in 1C-piRNAs eliminates any bias against 1C to frequencies that are expected by genomic origin (Fig. 4a insert). In contrast, purines (A and G) were not significantly altered in SL-mutants (Fig. 4c, d and Supplementary Fig. 4a, b). Interestingly, aversion of 1C due to repulsion by the SL has previously been suggested based on structural modeling and the inability to obtain such Argonaute-small RNA crystals[21,37]. The initial hypothesis of 1U-selection by attractive interactions, based on the visible fit of 1U in structural studies, might have diverted our attention from the less visible but potentially more significant negative selection of 1C: while 1U-enrichment by affinity requires two hydrogen bonds to stabilize the 1U-PIWI interaction, a single

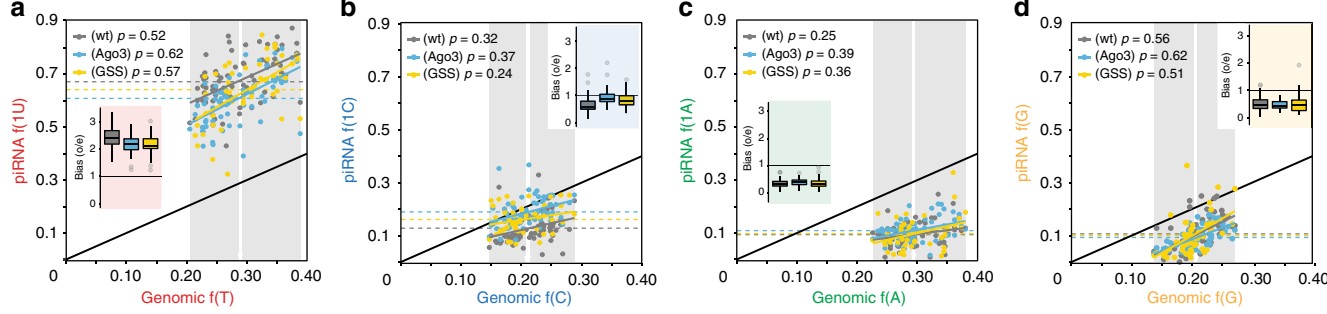

**Fig. 4** The 1U-bias of Piwi-piRNAs is established by a general mechanism that restricts all nucleotides but U in the first position. Piwi-piRNAs from OSC are grouped by piRNA generating clusters. The observed **a** 1U, **b** 1C, **c** 1A, and **d** 1G piRNAs (y-axis: piRNA f(1×)) are correlated with the expected genomic nucleotide frequencies (x-axis: genomic f(x)) of their generating cluster. Inserts represent the corresponding nucleotide bias (observed/expected nucleotide frequencies) as box plots with the median indicated as center line with the upper and lower quartile represented by the box. PiRNAs that are associated with FH-Piwi(wt) (gray) and two FH-Piwi-SL mutants, (Ago3) (blue) and (GSS) (yellow), are shown as median of three biological replicates (n = 3). [M = 1: unique mapping reads are considered]. (See also Supplementary Fig. 4 and Supplementary Data 6.)

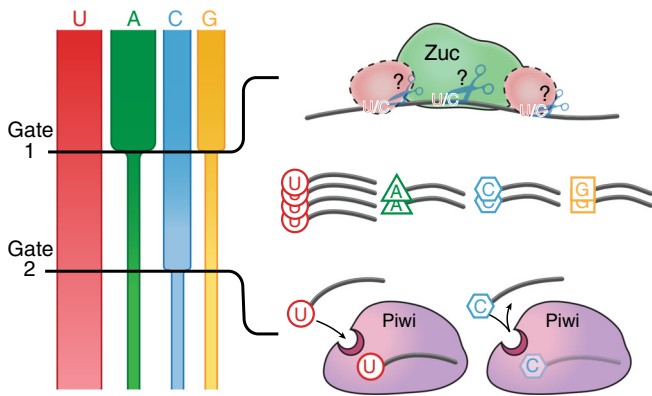

**Fig. 5** Model for the establishment of a 1U-bias. Our data support that the 1U-bias is established in two steps by differential gating against all nucleotides but Uridine. First, purines (A/G) are reduced during piRNA biogenesis in an SL-independent manner. Consecutively, 1C is discriminated against upon piRISC formation by interactions with Piwi's SL. (See also Supplementary Fig. 5.)

repulsive interaction is expected to reduce 1C binding (Fig. 5 and Supplementary Fig. 5). An exclusionary rather than affinity function of PIWI's SL offers an explanation as to why, in the only available structure of a PIWI protein (silkworm PIWI, i.e., SIWI), only one hydrogen bond is observed between the SL and 1U, despite the protein's marked preference for 1U piRNAs in vivo[33]. Taken together, our data suggest that the 1U-bias of Piwi-piRNAs is established through a two-step gating mechanism that sequentially reduces A, G, and C by SL-independent and SL-dependent preferences (Fig. 5).

## Discussion

The intricate establishment of the final 1U-bias inspires hypotheses about its role beyond reflecting a "key-lock fit" to stabilize piRISC. Binding preferences of silkworm PIWI in vitro[16], and the extensive interaction of human AGOs with the first U or A of small RNAs[21,30,31,34], might be a consequence rather than the cause of the 1U-bias, ensuring optimal RISC stability. This hypothesis could explain why positive interactions between the SL and the first base of the small RNA, requiring at least two hydrogen bonds, would favor both 1U and 1A equally, in contrast to the strong preference for 1U over 1A small RNAs

in vivo[21,34]. With the A/U-richness of piRNA precursors and the rejection of A and G during piRNA biogenesis, PIWI's SL only has to discriminate between 1U and 1C to reinforce an overall 1U preference in vivo.

But what is the main function of this 1U-bias and the subsequent restriction of piRNA sequence space? Reduction of 1G- and 1C-RNAs might decrease mis-incorporation of RNA fragments from protein coding sequences, and preferentially represent non-coding A/U-rich transcripts, including piRNA precursors. This effect might contribute to the observed preference of mRNA-derived piRNAs to originate from the A/U-rich 3′UTRs rather than G/C-rich coding sequences[29,43]. Furthermore, sequence preferences during piRNA processing are associated with relative U-depletion and G-enrichment in piRNA bodies, and directly modulate the sequence space for complementary target interactions. While we do not know the targeting rules of any nuclear piRISC, a mechanism following miRNA-like pairing[17] was recently identified for C. elegans PRG-1[18]. Such a seed-based mechanism involves base-pairing of positions two to eight of the piRNA with the target transcript, and additional 3′ supplemental interactions, all of which would be affected by the observed processing preferences.

Overall, the sequence bias during piRNA processing and piRISC formation could provide substrates for purifying selection in the ongoing arms race between genomes and their mobile parasites to improve specificity and efficiency of piRNA silencing, and avoid auto-aggression while restricting the entire transposome. As a defense against ancient retroviral invaders and other transposons, rules of piRNA silencing might foremost follow those that govern efficient viral defense: establishment of restriction and specificity at multiple independent steps to hamper development of resistance by the invader, where every contribution counts[44].

## Methods

**DNA constructs and transgenic flies**. Plasmids and oligonucleotides are listed in Supplementary Data 1. Piwi coding sequence (cds) was cloned into pENTRD/TOPO (ThermoFisher). Mutations in Piwi's specificity loop (SL) were introduced using the Q5 Site-Directed Mutagenesis Kit (NEB), and mutagenic primers were purchased as recommended by the manufacturer (NEBuilder). Expression vectors were generated through the LR Clonase (ThermoFisher) reaction, using the pPFHW destination vector (UASp promoter, N-terminal 3×FLAG/3×HA) (DGRC: #1125).

**CRISPR constructs for transgenic piwi>Gal4 flies**. For the sgRNA construct, oligonucleotides (piwi_sgRNA_f and piwi_sgRNA_r) were annealed and ligated into BbsI-digested U6-BbsI-chiRNA (Addgene # 45946). The donor construct for

genome-editing was generated as follows: pDONR-ETG3R and its homology arm-containing derivatives were cloned by HiFi Assembly (NEB). First, pDONR221 (ThermoFisher) was digested with PciI and EcoRI (NEB) and the KanR-containing 2.4 kb fragment was gel extracted and purified (Zymo Research). Next, EGFP-T2A-Gal4 was PCR amplified from burs-mCD8-EGFP-T2A-Gal4 (a gift of Dr. Benjamin White, Addgene plasmid #39463), and 3×P3-driven dsRed2 was PCR amplified from pT-GEM(1) (Addgene plasmid #62893). These three fragments were HiFi assembled to produce pDONR-ETG3R. Homology arms corresponding to 1 kb upstream and 1 kb downstream of the Piwi start codon were PCR amplified using genomic DNA from TH_attP2 nos-Cas9 flies (gift of Dr. Benjamin White) extracted using the GenElute Genomic DNA Miniprep Kit (Sigma-Aldrich). Finally, pDONR-ETG3R was digested with PciI and EcoRI (NEB), and the homology arms were assembled flanking the Gal4-containing cassette by HiFi Assembly (NEB). All oligonucleotides were purchased from IDT.

The CRISPR piwi>Gal4 allele was generated by co-injection of plasmids encoding the sgRNA and the donor construct for genome-editing into TH_attP2 nos-Cas9 flies (Bestgene plan Plan RI, Marker: RFP+/DsRed +). UAS-Piwi△SL transgenic flies were generated by random insertion into w1118 (Service type: Plan C; screening marker w+).

**Immunofluorescence and microscopy.** *Drosophila* ovaries fixed in 4% paraformaldehyde were dispersed into ovarioles to ensure maximum exposure to antibodies. Samples were washed 3 × 10 min in PBSTw (PBS+ 0.1% Tween) on rocker. Ovarioles were permeabilized for 30 min in PBSTr (PBS+ 0.2% Triton), washed, and blocked in 2% BSA for 1h with rocking. Primary antibody (α-HA mouse 1:100, Biolegend Cat#:90501; α-Piwi rabbit polyclonal antibody, immunized against (MADDQGRGRRRPLNEDC) 1:100) in 2% BSA was incubated overnight at 4 ˚C. Secondary antibodies coupled to Alexa fluorophores: goat α-mouse Alexa 488 (cat#: A-11029 1:500) and goat α-rabbit Alexa 488 (cat#: A-21245 1:500) in 2% BSA were incubated for 1 h, followed by a 30 min incubation with DAPI (1 μg/mL) and washed in PBSTw for 1 h. The stained ovarioles were mounted with VectaShield HardSet antifade mounting medium on Fisherbrand superfrost plus microscope slides with coverslip and edges sealed with nail polish. Images were taken using a Zeiss LSM710 confocal microscope (NIDDK microscopy core).

**Tissue culture of ovarian somatic sheath cells (OSC).** OSC were purchased from DGRC (OSS, stock #190) and cultured according to the DGRC protocol. Cells were transfected using Xfect (Clontech) according to the manufactures guidelines.

**Immunoprecipiation and small RNA library preparation for Illumina sequencing.** FLAG-HA-Piwi et and ΔSL constructs were immunoprecipitated using anti-FLAG M2 magnetic beads (Sigma-Aldrich). Lysis, immunoprecipitation and three washes were performed in IP-buffer (20 mM Tris HCl pH 7.5, 250 mM NaCl, 1% NP-40, 2 mM MgCl₂). NaCl was increased to 0.5 M for a fourth wash followed by a final rinse with IP buffer. RNA was extracted (Zymo Direct-zol RNA miniprep kit). Small RNA libraries for illumina sequencing were prepared according to Benhalevy et al.[45] with small modifications: in brief, libraries were size selected after 3′ ligation and 5′ linker-ligation. cDNA generated with Superscript IV reverse transcriptase was amplified with Q5 Polymerase (NEB) in a low cycle PCR (12 cycles). Pippin prep was used to remove ligated linker-linker (143–185 bp). Pilot PCR on 10% of pippin-prepped cDNA was performed to determine appropriate number of additional PCR cycles. Tapestation (Agilent) was used to determine quality and concentration of the resulting libraries. The samples were sequences on a HiSeq2500 sequencing system (Illumina) (NIDDK genomics core facility).

**Analyses of the next-generation sequencing data and code availability.** Mapping statistics are provided in Supplementary Data 2. Adapters and barcodes were removed using cutadapt (cutadapt 1.16). Structural RNAs were removed by mapping to a concatenated fasta file that includes (UCSC miRNA, rRNA, snRNA, snoRNA, tRNA) using STAR (STAR 2.5.4a)[46]. Unmapped reads were obtained and mapped to the *Drosophila* genome (dm6, igenomes) using STAR. Primary alignment for multimapping reads (M ≤ 100) were extracted using Flag (-F 256). Unique mappers (M = 1) were filtered with the Flag ("NH:i:1"). When both unique mappers and Multi mappers are included for our analyses, we call them All Mappers. PiRNA cluster intervals were defined according to the original definition: ≥1 unique mapping read per kilobase (kB) of genome space, over at least five kb[8]. Original and processed data files, and count tables are deposited in GEO (GSE115839). All the analyses and plotting was performed using R (version 3.5.0 or 3.3.1) and RStudio (Version 1.0.136) either locally or using National Institutes of Health High Performance Computing Cluster (http://hpc.nih.gov). The packages used in analysis were: data.table, plyr, dplyr, reshape, reshape2, Rmisc, GenomicRanges, GenomicAlignments, ShortRead, Rsamtools, Biostrings, rlist, ggplot2, ggrepel, scales, parallel, extrafont, BSGenome.Dmelanogaster.UCSC.dm6, and BSgenome.Mmusculus.UCSC.mm10[47].

**Generating count tables and plotting of data.** Count tables for each Figure are provided in Supplementary Data 3–6. The sample bam files were loaded into the R environment and all un-spliced reads in size range from 19 to 50-nt long were retained for the further analysis. Reads mapping to non-standard chromosomes

(Un_*) were removed. GenomicRanges R package was used to perform all the counting[48]. To count transposable elements (TEs), most recent rmsk.gtf file for dm6 genome was downloaded from TEtoolkit website[49]. To count cluster piRNAs, cluster.bed file was defined as described above.

To plot read size distribution, the individual sample read counts were normalized by total library size to the wt libraries. When replicates were available, we calculated and plotted mean and standard error values (Figs. 1d, 2c). To plot transposon targeting potential, reads mapping to TE antisense were retrieved, and annotated to transposon families according to repeat masker file (rmsk.gtf from TEtoolkit) (Figs. 1e, 2d). For plotting genomic annotation, reads mapping to both sense and antisense TEs, exons and introns were filtered respectively and annotated (Figs. 1f, 2e). Nucleotide frequency was calculated by counting identity of the first nucleotide of read (Figs. 1g, 2f). The genomic tracks were prepared in IGV by using a tiled data file (.tdf) that was derived from bam files using igvtools. Sequencing logos were generated with weblogo 3, for which fasta files of piRNA sequences were the primary requirement (Supplementary Figs. 1 & 2). For metagene analysis, first, genomic start position of every read was extended by 50-nucleotide upstream and downstream. Then, the sequences of corresponding genomic ranges were aggregated and the nucleotide abundance at each position was counted. To compare different nucleotides, individual nucleotide signatures were plotted on the same baseline using median expression over 101-nt window as baseline shift distance (Fig. 3a). To generate the background model, every nucleotide of piRNA-generating clusters was considered as potential piRNA, and the metagene was generated in the same manner as before (Fig. 3a insert). To simplify comparison of nucleotide signatures between samples, individual nucleotide frequencies were offset to avoid overlap, and correlation coefficients were calculated between wt and mutants (Fig. 3b–e). The same data were used to plot actual nucleotide frequencies and ratios of mutants compared to wt (Supplementary Fig. 3a–d). For calculating nucleotide bias, the cluster genomic nucleotide frequencies (expected) and the nucleotide frequencies of the 1st position of all reads (observed) were calculated. Then the ratio between observed (o) and expected (e) was designated as "nucleotide bias" (Fig. 4, Supplementary Fig. 4). To calculate correlations values, only complete pairwise observations were considered and Pearson correlation coefficients were calculated (Fig. 4). With the exception of size distribution plots, we only considered reads from 24 to 29-nt for all analyses and plotting. Fig. 4c is based on GSM2516689 (GSE95580).

**Molecular modeling.** Initial structural models of *Drosophila melanogaster* Piwi (from wildtype and mutant primary sequences) were generated using the Phyre2[50] protein fold recognition server in intensive mode. The top structures informing the *Dm* Piwi model were that of silkworm Piwi (Siwi)[33] and human Ago2[30]. The models output from Phyre2 then served as input for loop remodeling of the specificity loop sequences using Rosetta version 3.9[51]. Fifty structures were generated and scored for each mutant; the lowest energy structure for each sequence is depicted in Fig. 1 and Supplementary Fig. 5. The lowest energy structures were then superposed onto the structure of Siwi bound to RNA to assess the predicted interfaces of the specificity loops with the 5′ nucleotide. Molecular graphics were generated using PyMOL[52]. Electrostatic surfaces were calculated with APBS[53].

**Reporting summary.** Further information on experimental design is available in the Nature Research Reporting Summary linked to this article.

## Data availability
Original and processed data files, and count tables are deposited in GEO (GSE115839). All data is available from the authors upon reasonable request.

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

## Acknowledgements

We thank Jenny Hinshaw, Andy Kehr, Leo Kong, and Jinwei Zhang for advice on the design of the mutant specificity loops; Benjamin White's group (NIMH) for plasmids and fly strains; Markus Hafner for extensive discussions and for comments on the manuscript; We thank members of the Haase, Hafner and Mammen groups and the NIH RNA community, especially Aishe Angeletti Sarshad, Daniel Benhalevy, Dimitrios Anastasakis, Bob Crouch, Rich Maraia, Vivian Cheung, Nick Guydosh, and Pedro Batista for constructive discussions. L.J. is an investigator of the Howard Hughes Medical Institute. M. H. is a scholar of the Rita Allen Foundation. We also thank the NIH High-Performance Computing group, and the genomics and microscopy core facilities at the NIDDK. This work was supported by the intramural research program of the NIDDK.

## Author contributions

C.B.S. initiated the project and generated FH-Piwi-SL mutants, P.G. and S.M. analyzed Illumina sequencing data. A.R.E. and C.B.S. performed experiments in tissue culture and in flies with the help of L.B., S.S. and Y.S. J.J.I. generated select SL mutants and performed structural modeling experiments. M.H. advised bioinformatic and biostatistics analyses. L.J. advised structural design. A.D.H. conceived the project, designed experiments and wrote the manuscript. All authors commented on the manuscript.
