## [Peer Review File · Nature Communications]

Reviewers' comments:

Reviewer #1 (Remarks to the Author):

In this paper, the authors describe an interesting set of experiments analyzing the molecular mechanisms of 1U-bias of Piwi-piRNAs in OSCs and in Fly. Previous studies have proposed two independent hypotheses for the basis of 1U-bias. One is the Argonaute-MID domain-dependent and the other is Zuc-dependent mechanism. But it remains obscure which mechanism contributes to 1U-preference. Interestingly, the authors obtained a set of data that show the 1U-bias of Piwi-piRNAs is established by both MID-independent and MID-dependent mechanisms, that is called as two-step gating mechanism. The experiments in this paper are organized well and the story is clear. So, this reviewer recommends publication of this paper after the following points are revised.

Fig. 4c: The illustration shows that Zuc seems to make the same amount of 1C and 1U piRNAs at the Gate 1. Given that 1C-piRNAs of Piwi-SL(Ago3) increased "slightly" in fig.1g, Fig. 4c may confuse the readers. I recommend reducing the number of 1C-piRNAs in this figure.

Page 2, line 43: "abrogated" should be "abrogated".

Reviewer #2 (Remarks to the Author):

Stein and coworkers have addressed the 5' nucleotide bias of the Drosophila Piwi protein, through mutation of its so-called specificity loop (SL). These mutants show effects on 5'U bias, but still maintain a preference for U at the 5' position, as also found for the wild-type protein. They analyze the effects both in OSC cell culture as well as in flies.

I appreciate the experiments, and their outcomes are certainly interesting. However, I feel that the impact of the findings will be rather restricted to the Piwi-field, with perhaps interest from people working on RNA-protein interactions. But for the latter, I strongly feel the manuscript is too thin. Well considered, it only provides one experiment, where the SL of Piwi is mutated to different versions. Even though it is then characterized in cells and in flies, in essence this remains the same experiment. The authors analyze small RNA sequencing data to come to some interesting options, as to where the remaining bias comes from, but these options are not tested experimentally. Hence, this work is a rather preliminary set of data that may well develop into a really interesting story, but that remains in its current form largely correlative, and is too premature for a journal like Nature Comm.

The experiments as such appear sound, but I cannot start to sum up experiments needed to boost this work to a level expected for Nature Comm. The level of result I'd like to see though would be testing if the repulsion of 'C' by the SL is indeed all it takes to get the U bias; where does the bias against A/G come from? Does the Piwi SL lead to 'C' repulsion in AtAGO5)? Without significant steps into these directions I strongly feel that the advance that this work brings remains far from one would like to see.

Minor comment: the manuscript is written extremely densely and therefore sometimes a bit hard to follow. Also, at one point the authors suddenly talk about Piwi-LD, without explaining what it is. I could deduce it was 'loop-deficient' but for a reader this is far from convenient.

Reviewer #3 (Remarks to the Author):

Stein et al. describe an in-depth analysis of the structural components involved in the 1'U bias of piRNAs. A hallmark of piRNAs, in particular piRNAs generated by Zucchini-mediated processing, is

a strong enrichment of uridine in the first position of the piRNA. How this strong bias is established is, however not known and has been the source of speculation for over a decade, with the prevalent hypothesis being a structural selection by the SL loop of Piwi proteins. Thus a detailed analysis of the functional role of the SL loop in 5'U specification is an important and timely topic. In their study, Stein et al have identified that the 5'U bias is facilitated by the SL loop, likely via exclusion of 5'C-s by charge repel, but that the SL loop by itself is unable to establish the bias and that a mechanism independent of Piwi structure is responsible for biasing against purines. The data they provide is consistent with previously published results by the Siomi and Pillai labs and provides an elegant mechanism for establishing the nucleotide bias with a possible function in biasing against CG-rich protein coding sequences to avoid self-targeting.

I find the work thorough and well-executed and the topic well suited for publication in Nature Communications with some suggested additional analyses and changes to the text and figures.

Major comments:

- 1) Based on the noisiness of the data in the LD mutant (Fig S3) it seems like either protein levels or piRNA content are greatly reduced. It would be helpful to differentiate between these two possibilities by analyzing total protein content in transgenic flies using Western blot and by comparing piRNA fraction (normalized to miRNA).
- 2) The model put forward by the authors argues for a repulsive effect of charges in the SL loop that aid in reducing 5'C piRNAs in Piwi proteins. If possible, it would be nice to evaluate whether differences seen between the mutants in their ability to exclude 1'C (e.g. AtAgo5 with C-levels similar to WT and Aub with greatly increase 1'C content) correlates with the charge repel based on the modeling. A possible correlation would enhance the validity of their model.

Minor comments:

- 1) The text is oftentimes hard to comprehend and needs clarification
 - On line 89-90, I am not sure I understand how IP-s provide a view of Zuc products that are otherwise degraded. This sentence might need clarification.
 - Line 90 degraded "on" the absence should be degraded "in" the absence
 - The sentence starting on line 99 is hard to understand and could be formulated simpler.
 - Line 110 nt is missing after 101.
 - Line 129 Fig 4a-d. there is no panel d.
 - The authors should clarify what the take-home message of the sentence in lines 130-132 is. Does this mean that the U-bias is dependent on a process directly linked with the processing machinery or that it is not due to exchange during IP or something else?
 - In supplementary materials methods section the sentence "CRISPR constructs for transgenic piwi>Gal4 flies seems like it might have been meant as a sub-chapter, but if not then it is incomplete.
 - In supplementary materials on page 3 6lines from the top after "(12 cycles)" there should be a period or and "and".
- 2) Color choices (and line thicknesses) in the figures are aesthetically pleasing but are often not visible if viewed on a printout.
 - I suggest avoiding yellow lines (1D, 2C and throughout for GSS mutant labeling) and thin lines.
 - I suggest making sure that colors in the bar plots contrast well in b/w printouts. This would be especially important in figure 1G & 2F, where red (U) and blue (C) are indistinguishable.
 - In 1d, I find it hard to differentiate between Aub and Ago3 even in color on the screen as the two colors seem so similar.
 - The gray boxes in Figure 3 are invisible on my printout.
- 3) I wonder why the labels in figure 1D are in parentheses
- 4) In figure 1b construct number 8 is labeled Piwi N>P, while in 1D it shows as R>P. are these referring to different construct or is this a typo?
- 5) In either the Aub or Ago3 loop substitutions, as well as in 3 of the synthetic SL mutants there

seems to be a pronounced 1nt shift in piRNA size (fig 1d). Do the authors have any explanation for why this shift might occur despite the overall protein structure remaining unaltered?

6) In Fig S1, I wonder why the authors scaled the y axis to a max of 2 when all the letters are well below the 1 value. U bias might be better visible if the plots are re-scaled.

7) Based on figure S3 a&b it seem as if the U bias were stronger in the ovary than in OSC both at the 1nt position and in the surrounding area. Are germline clusters more U-rich than somatic clusters?

8) What additional information does 4b provide over 4a? my understanding is that it is simply dividing the values of the different mutants by those corresponding to the WT, or did I miss some additional content?

We wish to thank the reviewers for their comments and their suggestions to improve our manuscript. We hope that the reviewers will find all their concerns appropriately addressed in the revised version of the manuscript and the point-by-point response.

Below please find our detailed response (underlined and in italics) to the reviewers' comments (in bold). Corresponding changes in the manuscript and supplement are highlighted (yellow).

Reviewer #1 (Remarks to the Author):

In this paper, the authors describe an interesting set of experiments analyzing the molecular mechanisms of 1U-bias of Piwi-piRNAs in OSCs and in Fly. Previous studies have proposed two independent hypotheses for the basis of 1U-bias. One is the Argonaute-MID domain-dependent and the other is Zuc-dependent mechanism. But it remains obscure which mechanism contributes to 1U-preference. Interestingly, the authors obtained a set of data that show the 1U-bias of Piwi-piRNAs is established by both MID-independent and MID-dependent mechanisms, that is called as two-step gating mechanism. The experiments in this paper are organized well and the story is clear. So, this reviewer recommends publication of this paper after the following points are revised.

Fig. 4c: The illustration shows that Zuc seems to make the same amount of 1C and 1U piRNAs at the Gate 1. Given that 1C-piRNAs of Piwi-SL(Ago3) increased "slightly" in fig.1g, Fig. 4c may confuse the readers. I recommend reducing the number of 1C-piRNAs in this figure.

We thank the reviewer for this thoughtful comment! The model has been adjusted to reflect the enrichment for individual nucleotides in the first position and is now shown as Figure 5.

Page 2, line 43: "abrogated" should be "abrogated".

This error is corrected in the revised version of the text.

Reviewer #2 (Remarks to the Author):

Stein and coworkers have addressed the 5' nucleotide bias of the Drosophila Piwi protein, through mutation of its so-called specificity loop (SL). These mutants show effects on 5'U bias, but still maintain a preference for U at the 5' position, as also found for the wild-type protein. They analyze the effects both in OSC cell culture as well as in flies.

I appreciate the experiments, and their outcomes are certainly interesting. However, I feel that the impact of the findings will be rather restricted to the Piwi-field, with perhaps interest from people working on RNA-protein interactions. But for the latter, I strongly feel the manuscript is too thin. Well considered, it only

provides one experiment, where the SL of Piwi is mutated to different versions. Even though it is then characterized in cells and in flies, in essence this remains the same experiment. The authors analyze small RNA sequencing data to come to some interesting options, as to where the remaining bias comes from, but these options are not tested experimentally. Hence, this work is a rather preliminary set of data that may well develop into a really interesting story, but that remains in its current form largely correlative, and is too pre-mature for a journal like Nature Comm.

The experiments as such appear sound, but I cannot start to sum up experiments needed to boost this work to a level expected for Nature Comm. The level of result I'd like to see though would be testing if the repulsion of 'C' by the SL is indeed all it takes to get the U bias; where does the bias against A/G come from? Does the Piwi SL lead to 'C' repulsion in AtAGO5)? Without significant steps into these directions I strongly feel that the advance that this work brings remains far from one would like to see.

We are sorry for the misunderstanding of our data and hope that the revised version of the manuscript better communicates our message: Our data suggest that the bias against A/G is established upstream of Piwi-piRISC formation during piRNA biogenesis. This initial restriction is represented as the “first gate” in our model. Piwi’s SL contributes an additional selection against 1C pre-piRNAs (“second gate”), which is lost upon mutation of the SL. All mutants, including the substitution for the SL of AtAGO5, increase the association of Piwi with 1C-piRNAs (Fig. 1g and 2f) and 1C-piRNAs reach representation as expected by the genomic content of their cluster of origin in select mutants (Fig. 4b). According to suggestions by Reviewer #3 the electrostatics encountered by the first base upon binding are now modelled (Fig. S5).

Minor comment: the manuscript is written extremely densely and therefore sometimes a bit hard to follow. Also, at one point the authors suddenly talk about Piwi-LD, without explaining what it is. I could deduce it was 'loop-deficient' but for a reader this is far from convenient.

We hope that the revised version of the text has improved readability.

Reviewer #3 (Remarks to the Author):

Stein et al. describe an in-depth analysis of the structural components involved in the 1'U bias of piRNAs. A hallmark of piRNAs, in particular piRNAs generated by Zucchini-mediated processing, is a strong enrichment of uridine in the first position of the piRNA. How this strong bias is established is, however not known and has been the source of speculation for over a decade, with the prevalent hypothesis being a structural selection by the SL loop of Piwi proteins. Thus a

detailed analysis of the functional role of the SL loop in 5'U specification is an important and timely topic.

In their study, Stein et al have identified that the 5'U bias is facilitated by the SL loop, likely via exclusion of 5'C-s by charge repel, but that the SL loop by itself is unable to establish the bias and that a mechanism independent of Piwi structure is responsible for biasing against purines. The data they provide is consistent with previously published results by the Siomi and Pillai labs and provides an elegant mechanism for establishing the nucleotide bias with a possible function in biasing against CG-rich protein coding sequences to avoid self-targeting.

I find the work thorough and well-executed and the topic well suited for publication in Nature Communications with some suggested additional analyses and changes to the text and figures.

Major comments:

1) Based on the noisiness of the data in the LD mutant (Fig S3) it seems like either protein levels or piRNA content are greatly reduced. It would be helpful to differentiate between these two possibilities by analyzing total protein content in transgenic flies using Western blot and by comparing piRNA fraction (normalized to miRNA).

As the reviewer points out, our preparation of Piwi(LD)-associated piRNAs from both OSC and Dm ovaries reproducibly resulted in smaller illumina data sets compared to Piwi(wt), (GSS), or (Ago3) despite the same amount of biological starting material (OSC or ovaries). According to the reviewer's suggestion, we evaluated Piwi-LD protein levels and total small RNA profiles in mutant ovaries: FH-Piwi-LD protein levels are ~6-fold reduced compared to FH-Piwi(wt) when compared by Western blotting (a) and ~2.5 fold reduced compared by immunofluorescence (b) ([piwi]>Gal4/CyO; FH-Piwi-LD or -SL(wt)). In FH-Piwi-LD flies in a piwi null background ([piwi]>Gal4/[piwi]>Gal4; FH-Piwi-LD) total piRNAs levels were dramatically reduced compared to a FH-Piwi-SL(wt) rescue experiment ([piwi]>Gal4/[piwi]>Gal4; FH-Piwi-LD) (c). Taken together, our analyses show that both piRNA levels and Piwi protein levels are reduced in the LD mutant.

FH-Piwi-LD protein levels and -piRNAs are reduced. *FH-Piwi-LD and FH-Piwi-SL(wt) protein levels were quantified by (a) Western blotting and (b) immunofluorescence in ovaries heterozygous for [piwi]>Gal4 ([piwi]>Gal4/CyO; FH-Piwi-SL(wt) or -LD). (c) Nucleotide size distribution of total small RNAs (20-29 nt) from FH-Piwi-wt or -LD rescue experiments ([piwi]>Gal4/[piwi]>Gal4; FH-Piwi-SL(wt) or -LD). The reads count of uniquely mapping small RNAs [M=1] was normalized to annotated miRNAs. Insert: Small RNAs that originate from the extended flamenco region (ChrX: 21631259 – 22282926 (+))*

2) The model put forward by the authors argues for a repulsive effect of charges in the SL loop that aid in reducing 5' C piRNAs in Piwi proteins. If possible, it would be nice to evaluate whether differences seen between the mutants in their ability to exclude 1' C (e.g. AtAgo5 with C-levels similar to WT and Aub with greatly increase 1' C content) correlates with the charge repel based on the modeling. A possible correlation would enhance the validity of their model.

According to the reviewer's suggestion we performed a comprehensive modelling of the binding interfaces for Piwi's SL and SL mutants: A simplified depiction of potential interactions is integrated as Fig. S5b and electrostatic models as Fig. S5c-d in the revised version of the manuscript. Based on our modelling, none of the SL mutants retain their ability to disfavor 1C in the context of Piwi (Fig. S5d). These models agree with the observed increase in the 1C-piRNA fraction associated with all mutants in OSC (Fig. 1g) and in vivo (Fig. 2f and 4b). Replacing Piwi's SL with the SL of AtAgo5 increases the fraction of 1C-piRNAs to similar extent as other SL-replacements. With the precise positioning of the SL within Argonaute proteins, we believe that loss of specificity is easier to achieve than gain of specificity by introducing a foreign SL. We have not yet examined

the function of Aub's SL in the context of Aub and thus cannot comments on its contribution to the 1U bias of Aub-piRNAs in vivo. Predictions by Dr. Phil Zamore's group proposed that direct recognition of an A in the 10th position of the target strand could result in preferential production of 1U-piRNAs during ping-pong¹.

Minor comments:

1) The text is oftentimes hard to comprehend and needs clarification

We have revised the text to increase background and clarifications. We hope that the reviewer will find all points addressed appropriately:

- **On line 89-90, I am not sure I understand how IP-s provide a view of Zuc products that are otherwise degraded. This sentence might need clarification.**

The sentence has been changed to: "Our SL mutants eliminate SL-dependent nucleotide selection in the first position but retain the ability to associate with Zuc-dependent piRNAs. Thus, they likely grant an unrestricted view on piRNA 5'-termini generated by the Zuc-processor complex, which are otherwise rapidly degraded in the absence of their Piwi protein partner." (lines 130-134)

- **Line 90 degraded "on" the absence should be degraded "in" the absence**

This sentence has been corrected.

- **The sentence starting on line 99 is hard to understand and could be formulated simpler.**

The sentence has been changed to: "This pattern is reminiscent of a phased processing signature, and likely represent coupled processing of the 5' and 3' termini of neighboring piRNAs by a single Zuc-cleavage event." (line 159, 161)

- **Line 110 nt is missing after 101.**

This sentence has been corrected.

- **Line 129 Fig 4a-d. there is no panel d.**

This sentence has been corrected.

- **The authors should clarify what the take-home message of the sentence in lines 130-132 is. Does this mean that the U-bias is dependent on a process directly linked**

with the processing machinery or that it is not due to exchange during IP or something else?

This sentence has been removed, as the main message -that the 1U-bias is established during piRNA biogenesis- is already stated in the previous sentence.

• In supplementary materials methods section the sentence “CRISPR constructs for transgenic piwi>Gal4 flies seems like it might have been meant as a sub-chapter, but if not then it is incomplete.

This has been reformatted as a subheading.

• In supplementary materials on page 3 6lines from the top after “(12 cycles)” there should be a period or and “and”.

A period has been added.

2) Color choices (and line thicknesses) in the figures are aesthetically pleasing but are often not visible if viewed on a printout.

• I suggest avoiding yellow lines (1D, 2C and throughout for GSS mutant labeling) and thin lines.

According to the reviewer’s suggestion the line color has been changed to a darker shade that can be discriminated on a greyscale print out.

• I suggest making sure that colors in the bar plots contrast well in b/w printouts. This would be especially important in figure 1G & 2F, where red (U) and blue (C) are indistinguishable.

According to the reviewer’s suggestion, the shades and ordering of the colors have been adjusted to increase contrast between neighboring fields in Fig 1g and 2f. We also changed the order of the individual nucleotides to place C and U on opposing sites of the stacked bar, which increases overall contrast and provides a better visual evaluation of the changes in the fraction of 1C-piRNAs (related to the reviewer’s major comment #2).

• In 1d, I find it hard to differentiate between Aub and Ago3 even in color on the screen as the two colors seem so similar.

The colors have been changed according to the reviewer’s suggestion.

• The gray boxes in Figure 3 are invisible on my printout.

The shade of grey has been changed according to the reviewer’s suggestion.

3) I wonder why the labels in figure 1D are in parentheses

In Figure 1a and b we established the nomenclature for as FH-Piwi-SL(wt,...) with the type of SL substitution in parentheses.

4) In figure 1b construct number 8 is labeled Piwi N>P, while in 1D it shows as R>P. are these referring to different construct or is this a typo?

The typo in Fig. 1d has been corrected.

5) In either the Aub or Ago3 loop substitutions, as well as in 3 of the synthetic SL mutants there seems to be a pronounced 1nt shift in piRNA size (fig 1d). Do the authors have any explanation for why this shift might occur despite the overall protein structure remaining unaltered?

piRNA 3' termini are determined by the footprint of the associated PIWI protein and shaped by the nucleotide preferences of the processing machinery^{2,3}. The observed length distribution is a function of piRNA sequence diversity and not a distribution of 3' end diversity of individual piRNA sequences. Changes in sequence diversity could influence the distribution as well as changes in the overall structure of piRISC. While we are currently investigating this phenomenon in more detail, we choose the SL(Ago3) an (GSS) mutants for in-depth analysis partly for their unaltered piRNA length distribution (Fig. 2c).

6) In Fig S1, I wonder why the authors scaled the y axis to a max of 2 when all the letters are well below the 1 value. U bias might be better visible if the plots are re-scaled.

The scale has been changed to a max of 1 to increase visibility of the signatures as suggested by the reviewer.

7) Based on figure S3 a&b it seem as if the U bias were stronger in the ovary than in OSC both at the 1nt position and in the surrounding area. Are germline clusters more U-rich than somatic clusters?

The observed difference in overall U frequency is likely the result of differential representation of piRNA clusters in fly ovaries compared to OSC, as this difference is reduced when focusing on a region of flamenco that is similarly represented by corresponding piRNAs in each sample (chrX: 21704629-21819922).

Individual nucleotide frequencies acquired from metagene analysis for a 101 nt window surrounding the 5' end of Piwi-piRNAs (position 0) in OSC [n = 3] (a) and fly Ovary [n = 1] from a well-represented region of the flamenco piRNA cluster (chrX: 21704629-21819922). The number of uniquely mapping piRNA reads is indicated for the represented samples, 3 replicates for OSC (replicate #2) [replicate #3] and one sample for fly ovaries. The median 1U frequency (1U med) across the 101 nt window and the maximum 1U frequency (1U max) are indicated.

8) What additional information does 4b provide over 4a? my understanding is that it is simply dividing the values of the different mutants by those corresponding to the WT, or did I miss some additional content?

As the reviewer points out, Fig. 4b is just a simplification of the main message presented in Fig. 4a. The panel has been removed in the revised version. For clarity, we split the four panels from Fig 4a into panels a-d and moved the final model (previously Fig. 4c) into a new figure, Fig. 5. We would like to thank the reviewer for suggestion to extend our structural modelling of Piwi's SL and mutant SLs (now Fig. S5).

References:

1. Wang, W. *et al.* The Initial Uridine of Primary piRNAs Does Not Create the Tenth Adenine that Is the Hallmark of Secondary piRNAs. *Molecular Cell* **56**, 708–716 (2014).
2. Brennecke, J. *et al.* Discrete Small RNA-Generating Loci as Master Regulators of Transposon Activity in *Drosophila*. *Cell* **128**, 1089–1103 (2007).
3. Gainetdinov, I., Colpan, C., Arif, A., Cecchini, K. & Zamore, P. D. A Single Mechanism of Biogenesis, Initiated and Directed by PIWI Proteins, Explains piRNA Production in Most Animals. *Molecular Cell* **71**, 775–790.e5 (2018).

REVIEWERS' COMMENTS:

Reviewer #3 (Remarks to the Author):

the authors have sufficiently addressed my concerns and have improved the text and figures. Based on the quality of the data, the importance/impact of the findings and the additional data provided by the authors, I find the revised version to be well suited for publication in Nature Communications.

REVIEWERS' COMMENTS:

Reviewer #3 (Remarks to the Author):

the authors have sufficiently addressed my concerns and have improved the text and figures. Based on the quality of the data, the importance/impact of the findings and the additional data provided by the authors, I find the revised version to be well suited for publication in Nature Communications.

Author response: We wish to thank the reviewer for his/her help to improve the manuscript and for accepting our latest version!